# An Integrative Genomic Prediction Approach for Predicting Buffalo Milk Traits by Incorporating Related Cattle QTLs

**DOI:** 10.3390/genes13081430

**Published:** 2022-08-11

**Authors:** Xingjie Hao, Aixin Liang, Graham Plastow, Chunyan Zhang, Zhiquan Wang, Jiajia Liu, Angela Salzano, Bianca Gasparrini, Giuseppe Campanile, Shujun Zhang, Liguo Yang

**Affiliations:** 1Department of Epidemiology and Biostatistics, School of Public Health, Tongji Medical College, Huazhong University of Science and Technology, Wuhan 430030, China; 2Key Laboratory of Agricultural Animal Genetics, Breeding and Reproduction of Ministry of Education, Huazhong Agricultural University, Wuhan 430070, China; 3Livestock Gentec Center, Department of Agricultural, Food and Nutritional Science, University of Alberta, Edmonton, AB T6G 2C8, Canada; 4Department of Veterinary Medicine and Animal Productions, University of Naples “Federico II”, 80137 Naples, Italy

**Keywords:** buffalo, pGBLUP, genomic prediction, linear mixed model, enrichment, prior biological information

## Abstract

Background: The 90K Axiom Buffalo SNP Array is expected to improve and speed up various genomic analyses for the buffalo (*Bubalus bubalis*). Genomic prediction is an effective approach in animal breeding to improve selection and reduce costs. As buffalo genome research is lagging behind that of the cow and production records are also limited, genomic prediction performance will be relatively poor. To improve the genomic prediction in buffalo, we introduced a new approach (pGBLUP) for genomic prediction of six buffalo milk traits by incorporating QTL information from the cattle milk traits in order to help improve the prediction performance for buffalo. Results: In simulations, the pGBLUP could outperform BayesR and the GBLUP if the prior biological information (i.e., the known causal loci) was appropriate; otherwise, it performed slightly worse than BayesR and equal to or better than the GBLUP. In real data, the heritability of the buffalo genomic region corresponding to the cattle milk trait QTLs was enriched (fold of enrichment > 1) in four buffalo milk traits (FY270, MY270, PY270, and PM) when the EBV was used as the response variable. The DEBV as the response variable yielded more reliable genomic predictions than the traditional EBV, as has been shown by previous research. The performance of the three approaches (GBLUP, BayesR, and pGBLUP) did not vary greatly in this study, probably due to the limited sample size, incomplete prior biological information, and less artificial selection in buffalo. Conclusions: To our knowledge, this study is the first to apply genomic prediction to buffalo by incorporating prior biological information. The genomic prediction of buffalo traits can be further improved with a larger sample size, higher-density SNP chips, and more precise prior biological information.

## 1. Introduction

Genomic prediction is becoming increasingly important for animal and plant breeding programs because of its effectiveness in improving selection and reducing costs [1,2,3,4,5]. The application of genomic prediction to humans has also attracted substantial research interest in terms of human disease prevention and personalized medicine in the last decade [6,7,8]. Many existing genomic prediction methods rely on either linear mixed models (LMMs) or sparse regression models. Common examples include the genomic best linear unbiased predictor (GBLUP) [9,10,11,12] and Bayesian Alphabet methods [13,14,15,16]. LMMs and sparse regression models are based on almost diametrically opposed assumptions. Precisely, LMMs assume that all genetic variants have nonzero effect sizes and their effect sizes follow a normal distribution, whereas the sparse regression models assume that a relatively small proportion of variants affects the phenotype. Several methods considering a hybrid of the two assumptions have also been developed, and these methods combined the advantages of both LMMs and sparse regression models [17,18]. In addition, a machine-learning-based method named KAML was developed by taking full advantage of the efficient computing of LMMs and the accurate prediction of Bayesian methods [19]. However, most of these standard methods are based on statistical considerations and often ignore the prior knowledge of biological information, such as functional annotation, pathways, eQTL, and the known causal loci. Ignoring biological information is likely suboptimal, as studies have shown that incorporating the gene annotation coming from public databases or previous GWAS results can improve genomic prediction accuracy [20,21,22,23,24].

Our main application of interest is in genomic selection of buffalos (*Bubalus bubalis*), which is a key species for smallholder producers in developing countries (e.g., India, Pakistan, and China) [25,26,27,28,29,30] and an important milk resource for specialized markets. Parallel with the statistical methodological development, new SNP arrays for animal programs have also been developed. For example, recently, the 90K Axiom Buffalo SNP Array was designed and commercialized. Like other livestock high-density SNP chips, the 90K array for buffalo is expected to improve and speed up various genomic analyses, which include exploring genetic diversity, analyzing complex traits and diseases, and aiding genetic selection [25,26,27,28,29,30]. River-type buffalo have also been genetically selected for milk production and fertility traits in some countries by traditional methods. It has been shown that milk yield, milk components, and milk somatic cell counts have enough genetic variation for selection purposes [31,32,33,34,35,36,37]. Here, we ask an important question: Can we use genomic prediction to speed up the genetic gains for the buffalo population? While there are different buffalo production systems around the world, the production records are particularly limited, especially when compared to those of cow. In addition, buffalo genome research is lagging behind cow genome research, and existing approaches have to align buffalo SNPs to the bovine genome for further study [28]. Therefore, we ask the second question: Can we improve the genomic prediction performance for buffalo milk traits with the limited sample size by incorporating the related cattle QTLs?

Motivated by both methodological interest and the above two application questions, we propose a statistical approach to incorporate prior biological information in the widely used genomic best linear unbiased predictor (pGBLUP) to improve genomic prediction. We first simulated several scenarios to test the stability and advantages of the approach. Then, we applied the approach for the genomic prediction of buffalo milk traits by incorporating the known cattle milk trait QTL information from the animal QTL database [38].

## 2. Materials and Methods

### 2.1. Statistical Model

The basic idea behind our approach is to fit the effect sizes of all SNPs as random effects relying on an LMM framework. We divided the SNPs into two groups based on a priori biological information, and we assumed that these two groups of SNPs have different effect sizes:(1)$$y=\mathrm{Zu}+Z_{1}u_{1}+\epsilon ,$$
where y is an n by 1 phenotype vector, which has been standardized to have mean 0 and variance 1 to remove the intercept in the equation, Z is an n by m genotype matrix for all SNPs, $Z_{1}$ is an n by p genotype matrix for p SNPs that are part of Z, genotype matrix Z and $Z_{1}$ were standardized as suggested [39], u is an m by 1 vector of small effect sizes for all SNPs, $u_{1}$ is a p by 1 vector of additional effect sizes for p selected SNPs, $u\sim N\left(0,\frac{\sigma_{\mathrm{small}}^{2}}{m}\right),{\;u}_{1}\sim N\left(0,\frac{\sigma_{\mathrm{large}}^{2}}{P}\right),\epsilon \sim N\left(0,{\;\sigma }_{e}^{2}\right)$. It should be noted that the SNPs in $Z_{1}$ have both large effects and small effects, which can be drawn from $N\left(0,\;\left(\frac{\sigma_{\mathrm{small}}^{2}}{m}+\frac{\sigma_{\mathrm{large}}^{2}}{P}\right)\right)$ [10,17,18]. Recent studies found that some regions or genes in the genome were heritability enriched for complex traits and disease [9,40,41,42,43,44]. In this model, we extracted the biologically functional information (pathway annotation, specific gene expression, and GWAS loci information) from a public database (e.g., https://www.animalgenome.org/ (accessed on 10 October 2017)) to serve as priors to determine which set of SNPs belongs to $Z_{1}$. To determine whether the “prior information” is meaningful and promising, the fold of enrichment (fe) was used as in the stratified LD score regression or MQS [42,43]:(2)$$\;\;\;\;\;\;\;\;fe=\left(\sigma_{\mathrm{large}}^{2}/\left(\sigma_{\mathrm{large}}^{2}+\sigma_{\mathrm{small}}^{2}\right)\right)/\left(p/m\right)+1.$$

When fe>1, the genome region based on “prior information” is suggested to be heritability enriched for the complex traits and disease, and SNPs with biological information will tend to have a large effect size, while SNPs without biological information will tend to have a small effect size [42,43]. Different from the Bayesian Alphabet methods [13,14,15,16,20], which use a time-consuming MCMC algorithm to determine the SNP effect distribution, our approach directly designs the SNP effect distribution based on the prior biological information. We name this approach as the incorporating prior biological information in genomic best linear unbiased predictor (pGBLUP). Equation (1) can be written as:(3)$$y=g_{s}+g_{l}+\epsilon ,$$
where $g_{s}=\mathrm{Zu}$, $g_{l}=Z_{1}u_{1}$, $g_{s}\sim MVN\left(0,\;A\sigma_{\mathrm{small}}^{2}\right)\;$,${\;g}_{l}\;\sim MVN\left(0,\;A_{1}\sigma_{\mathrm{large}}^{2}\right)$, *MVN* denotes the multivariate normal distribution, and A and $A_{1}$ are the realized genetic relationship matrix (GRM) [39]. $\sigma_{\mathrm{small}}^{2}$,$\;\sigma_{\mathrm{large}}^{2}$, and $\sigma_{e}^{2}$ are estimated firstly. Then, $g_{s}$ and $g_{l}$ can be estimated as:(4)$${\hat{g}}_{s}=A{\hat{\sigma }}_{\mathrm{small}}V^{-1}y,\;{\hat{g}}_{l}=A_{1}{\hat{\sigma }}_{\mathrm{large}}V^{-1}y,$$
where $V=A{\hat{\sigma }}_{s}^{2}+A_{1}{\hat{\sigma }}_{1}^{2}+I{\hat{\sigma }}_{e}^{2}$, I is an n by n identity matrix, and the SNP effect sizes can be estimated as
(5)$$\hat{u}=Z^{T}{\left(ZZ^{T}\right)}^{-1}{\hat{g}}_{s},\;{\hat{u}}_{1}=Z_{1}^{T}{\left(Z_{1}Z_{1}^{T}\right)}^{-1}{\hat{g}}_{l}.$$

In the training dataset, about 80% of all individuals in the simulations, the SNP effects $\hat{u_{1}}$ and $\hat{u}$ are estimated jointly. In the test dataset, about 20% of all individuals, the phenotypes can be predicted as:(6)$$\hat{y}=Z^{\mathrm{test}}\hat{u}+Z_{1}^{\mathrm{test}}{\hat{u}}_{1}$$
where $Z^{\mathrm{test}}$ is the standardized genotype matrix for all SNPs, $Z_{1}^{\mathrm{test}}$ is the standardized genotype matrix for the p SNPs in the test population, and $\hat{y}$ is the predicted values, known as the genomic estimated breeding value (GEBV).

### 2.2. Animal Resources and Genomic Information

German Holstein genomic prediction population [22,45]: The genotype data consisted of 5024 samples and 42,551 SNPs after removing SNPs that had a Hardy–Weinberg equilibrium (HWE) *p*-value < 10^−4^, genotype call rate < 95%, or minor allele frequency (MAF) < 0.01. All SNP positions were re-coded by the provider for confidentiality, and the genotypes of the population were used to simulate the phenotype in this study.

Water buffalo data [46]: The genotype data consisted of 412 Italian Mediterranean buffaloes, which were genotyped by the 90K Axiom Buffalo SNP Array. Then, 60,387 SNPs were retained after removing SNPs that had an HWE *p*-value < 10^−5^, genotype call rate < 97%, or MAF < 0.05. Six buffalo milk traits (peak milk yield (PM), total milk yield (MY), fat yield (FY), fat percentage (FP), protein yield (PY), and protein percentage (PP)) were recorded and adjusted to 270 days in milk, as suggested by [47]. The estimated breeding value (EBV) for the six traits was estimated with a univariate animal model using ASReml 3.0 [48]. The deregressed EBV (DEBV) of the six milk production traits was calculated according to [49]. The details of the data processing were described by Liu et al. [46]. Both the EBV and DEBV were used as the phenotype in this study.

Cattle milk QTLs: The QTLs of 112 cattle-milk-related traits were downloaded from the animal QTL database (https://www.animalgenome.org/cgi-bin/QTLdb/index accessed on 10 October 2017) during October 2017; those traits included milk yield, milk fat, milk protein, and some other milk component traits. Based on the Bos taurus UMD3.1 genomic assembly, the genes within 50 kb of the QTL regions were selected as genes associated with the trait. We only focused on the QTL regions with a length smaller than 40kb, which had the largest proportion of QTLs. After initial filtering, 2435 genes were selected to be associated with the milk traits, including some genes associated with several milk traits. Then, we only kept 396 genes (see Appendix A) associated with at least four traits as the prior biological information for the following real data application study for the genomic prediction of buffalo milk traits.

### 2.3. Simulations

We used the real genotypes of the German Holstein genomic prediction population to simulate the phenotypes with the following steps:(1)Set the causal segments: The genotype matrix was standardized, and the 42,551 SNPs were divided into 1000 approximately equally sized segments, with 42 or 43 SNPs in each segment; *s* (10/25/50/100/500) segments were randomly selected as causal segments in our simulation settings, and the 10 SNPs in the center of each segment were then selected as causal SNPs; thus, the total number of causal SNPs (k) was 100/250/500/1000/5000, while the total number of SNPs in the causal segments was p≈s∗42.5.(2)Simulate the SNP effects and phenotype: Firstly, all SNPs were simulated with the small effects following a normal distribution N(0, 0.25/42,551); the k causal SNPs were simulated with additional effects following a normal distribution N(0, 0.25/k). Then, the residual errors were sampled from a normal distribution N(0, 0.5), so that the total heritability of the simulated trait was 0.5. Based on Equation (1), for each individual, the phenotype was obtained as the summation of small effects, large effects, and the residual error.(3)Five-fold cross-validation: The 5024 individuals were divided into five groups, with 1004 or 1005 individuals in each group. Each time, one group of individuals was set as the test dataset, while the rest of the groups of individuals were set as the training dataset (i.e., five-fold cross-validation). We applied the pGBLUP approach in two ways to predict the performance in the test dataset: only the SNPs in the causal segments were set in $Z_{1}$; SNPs in both the causal segments and non-causal segments were selected in $Z_{1}$. We also applied the traditional GBLUP method [3] and the BayesR method [16] to compare the performance. The GBLUP method assumes the effect size for every variant is sampled from the same normal distribution; the BayesR method uses an MCMC algorithm to estimate variant effects, which are modelled as a mixture distribution of four normal distributions, including a null distribution, $N\left(0,\;0.0\sigma_{g}^{2}\right)$, and three others: $N\left(0,\;0.0001\sigma_{g}^{2}\right)$, $N\left(0,\;0.001\sigma_{g}^{2}\right)$, and $N\left(0,\;0.01\sigma_{g}^{2}\right)$, where $\sigma_{g}^{2}$ is the additive genetic variance for the trait.

### 2.4. Genomic Prediction of Buffalo Milk Traits

As the draft genomic sequence of the buffalo is currently not assigned to chromosomes, the chromosome and position for all SNPs in the 90K Axiom Buffalo SNP Array were based on the bovine UMD 3.1 genome sequence [28]. This also facilitated the use of the bovine gene annotation information; 1279 SNPs were selected within 10 kb of the 396 cattle-milk-trait-associated genes and set in $Z_{1}$ of the pGBLUP model. The fe of “prior information” for each trait was estimated using all individuals. Then, we applied three methods, the GBLUP, BayesR, and pGBLUP, to perform the genomic prediction of the six buffalo milk traits with five-fold cross-validation: the 412 individuals were divided into five groups, with 82 or 83 individuals in each group; each time, one group of individuals was set as the test dataset, while the rest of the groups of individuals were set as the training dataset.

### 2.5. Computation

For the GBLUP and pGBLUP, we used the GCTA [39] to perform REML, fe estimation, and the BLUP (http://cnsgenomics.com/software/gcta/#Download (accessed on 10 October 2017)). For BayesR, we used BayesR with the default parameters (https://github.com/syntheke/BayesR (accessed on 10 October 2017)). For the data cleaning and processing, PLINK (https://www.cog-genomics.org/plink2 (accessed in October 2017)) [50] and R (https://www.r-project.org/ (accessed on 10 October 2017)) [51] were applied.

## 3. Results

### 3.1. Predictive Accuracy in Simulations

In our simulation studies, the causal regions were simulated and known. Firstly, we assessed the performance of different methods when only the SNPs in the causal segments were set in $Z_{1}$. Based on the correlation (0.45~0.56) between the predicted values and the simulated values in the test dataset, BayesR and the pGBLUP greatly outperformed the GBLUP in the sparse simulation settings (i.e., only 10 causal segments) (Figure 1A,B). If the simulated SNP effects tended to be polygenic (i.e., 500 causal segments), the three methods had a similar performance, and the GBLUP even slightly outperformed BayesR (Figure 1A,B). It should be noted that the pGBLUP had the best predictive performance in all simulation settings, especially in sparse settings in which only several causal segments greatly affected the phenotype and the causal segments were accounted for in $Z_{1}$. The performance of the pGBLUP depended on the fe of the SNP set in $Z_{1}$; when the fe decreased (Figure 1C), the predictive accuracy gain compared with the GBLUP would be reduced from about 20% to 1% (Figure 1B). The fe estimations were centered on the truth at the median simulated fe, while they tended to be underestimated at the large simulated fe (Figure 1C).

Then, we assessed the performance of the pGBLUP when SNPs in both the causal segments and non-causal segments were set in $Z_{1}$ to examine the predictability of our method. We selected 430 SNPs from different numbers of causal segments and non-causal segments and set them in $Z_{1}$ when simulating 10 causal segments (Figure 2). When all SNPs in $Z_{1}$ were from the non-causal segments, the pGBLUP and GBLUP had similar performance. Upon increasing the number of SNPs in $Z_{1}$ from causal segments, the predictability of the pGBLUP became better and better (Figure 2A) and the fe of the SNPs set in $Z_{1}$ also became larger and larger (Figure 2B). The pGBLUP would have the best performance when all and only the causal segments were accounted for in $Z_{1}$.

### 3.2. Genomic Prediction of Buffalo Milk Traits

In our real data application studies, according to the prior biological information of 396 cattle-milk-trait-associated genes, 1279 SNPs within 10 kb of those genes were set in $Z_{1}$ of the pGBLUP model. The fe estimations of the selected 1279 SNPs in $Z_{1}$ are shown in Figure 3A. When the DEBV was regarded as the phenotype, there was no obvious enrichment for the selected SNPs, and the 1279 SNPs had fe close to 1 as other SNPs. When the EBV was regarded as the phenotype, the 1279 SNPs had a small fe for traits FY270, MY270, PY270, and PM.

Three methods were applied for genomic prediction of buffalo milk traits using the GEBV and EBV as the phenotypes (Table 1). The heritability estimations of the buffalo milk traits ranged from 0.702 to 0.793 using the DEBV as the phenotype and from 0.599 to 0.741 using the EBV as the phenotype. The correlations between the GEBV and DEBV ranged from 0.304 to 0.442, while the correlations between the GEBV and EBV ranged from 0.180 to 0.398. The three methods had similar performance for genomic prediction with large standard errors, and using the DEBV as the phenotype had better genomic prediction performance than using the EBV as the phenotype on average (Figure 3B and Table 1).

## 4. Discussion

We proposed a new genomic prediction approach called the pGBLUP, which incorporates prior biological information in the LMM. Several methods incorporating prior biological information in the LMM were also developed recently using different strategies from ours: The BLUP|GA [21,22,52,53] and single-step GBLUP accounting for causative quantitative trait nucleotides (QTNs) [24] model one weighted trait-specific GRM based on the prior biological information; CVAT [23,54,55] models two genetic variances in the LMM, while the SNPs in the two genetic components should be disjointly divided by the prior biological information. Both the BLUP|GA and CVAT selected and tested prior biological information using some iteration or permutation procedures, while we followed the main idea from [42,43] and directly used the heritability enrichment (fe) to measure the importance of the prior biological information. As our simulation results show in Figure 1 and Figure 2, the predictability of pGBLUP could be improved as the fe of prior biological information increased. Under the Bayesian framework, the extension of BayesR, BayesRC, was introduced, which incorporates prior biological information in the analysis by defining classes of variants likely to be enriched for SNPs with prior biological information, which showed competitive performance in the QTL mapping and genomic predictions [20].

In our simulations, the real genotypes of the cattle were divided into 1000 segments without considering the linkage disequilibrium (LD), which is not ideal. However, this should not affect our simulation purpose of illustrating the relationship between the predictive performance and fe; the pGBLUP and BayesR performed better with larger fe, which indicated less genes had large effects on the traits, and the GBLUP performed better with smaller fe, indicating more genes had small effects on the traits (Figure 1). While ignoring the LD in the simulations may affect the fe estimation, when there were 10 causal segments, the fe of all SNPs in the 10 causal segments was underestimated (Figure 1C). The missing fe were shared by the nearby non-causal segments, which could be observed for the fe overestimation when all SNPs from the non-causal segments had fe>1 (Figure 2B). In summary, the fe estimation was a good measurement for the importance and quality of the prior biological information. If the prior biological information was appropriate, the pGBLUP would outperform BayesR and the GBLUP; otherwise, it performed slightly worse than BayesR and equal to or better than the GBLUP (Figure 2A).

We applied the pGBLUP to our published buffalo data [46] with another two popular genomic prediction methods: the GBLUP and BayesR. Due to the delayed buffalo genome research, prior biological information is very rare. For buffalo traits, some cattle-related QTLs were also identified in buffalo [25,28,56,57,58,59], while the sample size for the buffalo population was too small to detect more causal variants. In this study, we incorporated the known cattle milk trait QTLs [38] for genomic prediction of buffalo milk traits in the pGBLUP. The prior biological information borrowed from cattle showed median enrichment for FY270, MY270, PM, and PY270 using the EBV as the phenotype. If the DEBV was used as the phenotype, the prior biological information only showed small enrichment for FY270, which had the largest enrichment when using the EBV as the phenotype (Figure 3). The fe estimations suggested that the prior information has the potential to improve the genomic predictability for FY270, MY270, PM, and PY270 if the EBV was used as the phenotype and for FY270 if the DEBV was used as the phenotype. The predictabilities of the three methods did not vary much (Table 1); BayesR and the pGBLUP did not show an advantage in genomic prediction, indicating the buffalo milk traits were less artificially selected for genes with large effects compared with cattle [13,16,52]. The other reasons for the small difference may be due to the limited sample size of the buffalo population in this study, which was indicated by the large standard errors of the correlations. In addition, we used a relatively loose threshold for the HWE test (*p*-value < 10^−5^) to remove the variants due to the genotyping errors, which was also likely to remove the causal variants under selection, thus affecting the performance of the genomic prediction. The heritability estimations using the DEBV and EBV as the phenotypes were larger than those using the original records directly [37,60,61], because some environmental effects were already removed for the DEBV and EBV. As previous research, we also noticed that the DEBV as the response variable yielded more reliable genomic predictions than the traditional EBV [49,62]. When the DEBV was used as the phenotype, the heritability estimations could reach 0.793 for PY270; the maximum achievable correlation between the predicted and observed traits was $\sqrt{0.793}=0.891$, but the mean correlations using the GBLUP, pGBLUP, and BayesR were 0.442, 0.439, and 0.439 (Table 1), so there was a large gap between the maximum achievable correlation and the real correlation. Based on the simulation results that the simulated heritability was 0.5 and the maximum achievable correlation was $\sqrt{0.5}=0.707$, the realistic correlation could reach 0.58 (Figure 1A); we believe that the genomic prediction of buffalo traits can be further improved with a larger sample size, higher-density SNP chips, and more precise prior biological information.

## 5. Conclusions

We proposed a genomic prediction approach, the pGBLUP, which has the potential to improve the genomic prediction performance by incorporating the proper prior biological information. The pGBLUP uses heritability enrichment to quickly check the importance of the prior biological information. We also applied the pGBLUP to incorporate the milk-related QTL information from cows for genomic prediction of buffalo milk traits. We found that some cattle-milk-related QTLs also played an important role in buffalo milk production traits. We believe that genomic prediction of buffalo traits can be further improved with a larger sample size, higher-density SNP chips, and more precise prior biological information.

## Figures and Tables

**Figure 1 genes-13-01430-f001:**
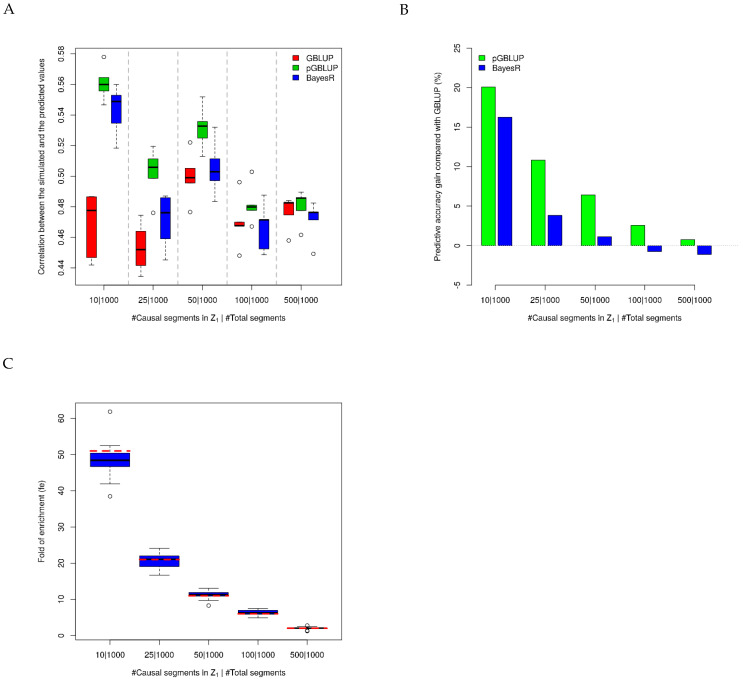
Simulation results when only the SNPs in the causal segments are set in $Z_{1}$. (**A**) The correlation between the predicted values and the simulated values in the test dataset using three methods in different simulation settings. (**B**) The percentage of predictive accuracy gain for the pGBLUP and BayesR compared with the GBLUP. (**C**) Fold of enrichment (fe) estimations using all individuals with 20 replicates; the red dashed lines represent the true values (from left to right: 51, 21, 11, 6, and 2). The black solid lines in (**A**,**C**) represent the median values of the estimations.

**Figure 2 genes-13-01430-f002:**
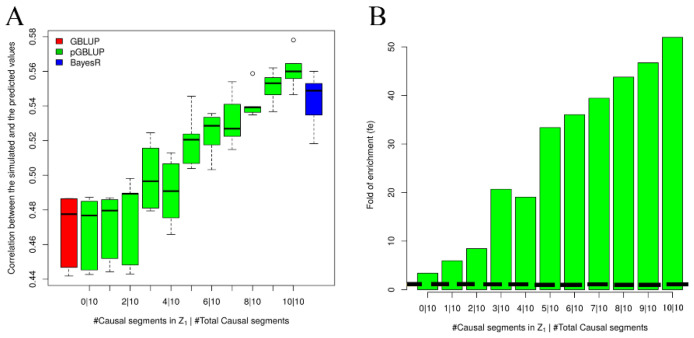
Simulation results when t = 0\1\2\3\4\5\6\7\8\9\10 causal segments and (10-t) non-causal segments are set in $Z_{1}$ at 10 simulated causal segments. (**A**) The correlation between the predicted values and the simulated values in the test dataset. (**B**) Average fold of enrichment (fe) estimations using the training dataset in the five-fold cross-validation; the black dashed line is fe=1.

**Figure 3 genes-13-01430-f003:**
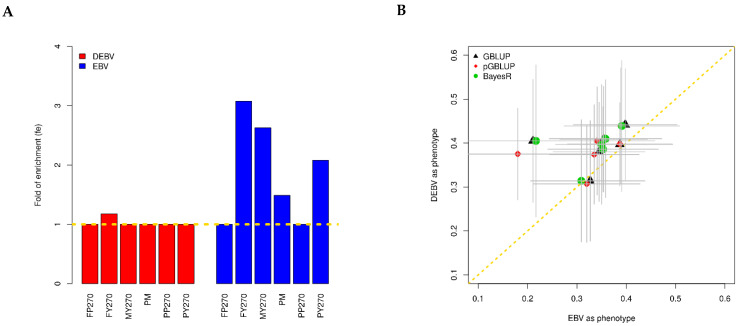
Genomic prediction results for buffalo milk traits. (**A**) Fold of enrichment (fe) of the selected 1279 SNPs in $Z_{1}$ using the DEBV and EBV as the phenotypes; the gold dashed line is fe=1. (**B**) Genomic prediction performance using the EBV and DEBV as the phenotypes; the points represent the mean correlations between the GEBV and EBV (or DEBV), and the lines represent the standard errors. PM: peak milk yield; MY270: 270-day total milk yield; FY270: 270-day fat yield; FP270 = 270-day fat percentage; PY270: 270-day protein yield; PP270: 270-day protein percentage.

**Table 1 genes-13-01430-t001:** Genomic prediction results of buffalo milk traits using the GEBV and EBV as the phenotypes. The mean and standard error of the correlation between the predicted and the true values in the five-fold cross-validation are reported. PM: peak milk yield; MY270: 270-day total milk yield; FY270: 270-day fat yield; FP270 = 270-day fat percentage; PY270: 270-day protein yield; PP270: 270-day protein percentage.

	Trait	h^2^	GBLUP	pGBLUP	BayesR
DEBV	FP270	0.713 ± 0.112	0.314 ± 0.137	0.307 ± 0.133	0.314 ± 0.139
	FY270	0.703 ± 0.119	0.38 ± 0.113	0.374 ± 0.113	0.397 ± 0.136
	MY270	0.753 ± 0.112	0.409 ± 0.12	0.405 ± 0.123	0.41 ± 0.134
	PM	0.702 ± 0.115	0.405 ± 0.14	0.375 ± 0.104	0.405 ± 0.173
	PP270	0.75 ± 0.112	0.397 ± 0.095	0.398 ± 0.092	0.386 ± 0.097
	PY270	0.793 ± 0.108	0.442 ± 0.127	0.439 ± 0.132	0.439 ± 0.149
EBV	FP270	0.741 ± 0.114	0.327 ± 0.111	0.32 ± 0.108	0.309 ± 0.103
	FY270	0.631 ± 0.124	0.345 ± 0.093	0.335 ± 0.091	0.35 ± 0.093
	MY270	0.658 ± 0.122	0.354 ± 0.089	0.341 ± 0.077	0.358 ± 0.114
	PM	0.599 ± 0.123	0.211 ± 0.22	0.18 ± 0.167	0.217 ± 0.241
	PP270	0.726 ± 0.115	0.387 ± 0.107	0.387 ± 0.106	0.353 ± 0.112
	PY270	0.738 ± 0.116	0.398 ± 0.105	0.389 ± 0.101	0.391 ± 0.117

## Data Availability

The production data of buffalo used in this study were provided by The Italian Buffalo Breeders Association (ANASB), which is responsible for the official herd book of the buffalo population in Italy. The buffalo data are available from the authors upon reasonable request. The cattle data were provided by Zhe Zhang from the South China Agricultural University (zhezhang@scau.edu.cn), which are available from Zhe Zhang upon reasonable request.

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
