# Peer review of "An Integrative Genomic Prediction Approach for Predicting Buffalo Milk Traits by Incorporating Related Cattle QTLs"

_genes, 2022, doi:10.3390/genes13081430_

Round 1
Reviewer 1 Report
Abstract
Line 16-20: The objective of the justice is lacking in the abstract.
Line 23: Please be specific as regards ‘prior biological information’.
Line 24: There is a need to provide a range of the heritability values.
Introduction
Line 54: Please expatiate on ‘knowledge of biological information’.
Line 59: Give examples of such developing countries and provide the source(s).
Materials and Methods
Line 100: Indicate the database.
Line 103-105: Is ‘?? > 1’ a threshold? Please provide a reference.
Line 117-118: Is there any ratio between the training and testing data sets?
Line 131: Change ‘trait’ to ‘traits’.
Results
Line 196: Provide the range of correlation coefficients.
Line 205: Provide the range of predictive accuracy.
Discussion
The Discussion is fairly okay considering the fact that the study is relatively new.
Conclusions
Line 316: Provide the cattle milk related QTL.
References
All the 62 references are in the list.
Table S1
Why CHR X for gene MBNL3- WWC3?
Author Response
Response of Comments and Suggestions for Reviewer 1
Abstract
Line 16-20: The objective of the justice is lacking in the abstract.
Response: Thanks for your suggestion. We have revised the objective as “To improve the genomic prediction in buffalo, we introduced a new approach (pGBLUP) for genomic prediction of six buffalo milk traits by incorporating QTL information for cattle milk traits in order to help improve the prediction performance in buffalo” in the abstract.
Line 23: Please be specific as regards ‘prior biological information’.
Response: Thanks for your suggestion. In our simulation, prior biological information was the known causal loci, so we specified the prior biological information (i.e. the known causal loci) in the abstract.
Line 24: There is a need to provide a range of the heritability values.
Response: Thanks for your comment. In our real data application, we did not focus on the heritability estimation, and we were interested in the heritability enrichment analysis (equation 2 in the main text) instead. Based on the definition of fold of enrichment (fe), we found that the heritability of buffalo genomic region corresponding to the cattle milk traits QTLs was enriched, which suggested that the SNPs in buffalo genomic region corresponding to the cattle milk traits QTLs might have large effects. Although we did not estimate and provide the heritability, we added the fold of enrichment (fe) in the abstract.
Introduction
Line 54: Please expatiate on ‘knowledge of biological information’.
Response: Thanks for your suggestion. We have revised the sentence as “... ignore the prior knowledge of biological information, such as functional annotation, pathways, eQTL and the known causal loci.” in the revised introduction.
Line 59: Give examples of such developing countries and provide the source(s).
Response: Thanks for your suggestion. We have revised the sentence as “... key species for smallholder producers in developing countries (e.g. India, Pakistan, and China)” in the revised introduction.
Materials and Methods
Line 100: Indicate the database.
Response: Thanks for your suggestion. We have added the database as “... from public database (e.g. https://www.animalgenome.org/) to serve as priors to determine which set of SNPs...” in the materials and methods.
Line 103-105: Is ‘?? > 1’ a threshold? Please provide a reference.
Response: Thanks for your comment. The fe is the fold of enrichment in heritability enrichment analysis. When fe > 1, it means that SNPs in the corresponding functional region might have large effects, and it is a commonly used threshold in previous studies. We added the references in the section of materials and methods.
Line 117-118: Is there any ratio between the training and testing data sets?
Response: Thanks for your comment. We used five-fold cross validation in our simulations, so the ratio between the training and testing data sets was 4. We have revised the sentence as “In the training data set, about 80% of all individuals in the simulations.... In the testing data set, about 20% of all individuals....” in the materials and methods.
Line 131: Change ‘trait’ to ‘traits’.
Response: Thanks for you pointing this typo. We have revised it.
Results
Line 196: Provide the range of correlation coefficients.
Response: Thanks for your suggestion. The range of correlation coefficients was shown in figure 1A. We also revised the sentence as “Based on correlation (0.45~0.56) between the predicted values and the simulated values in the testing data set, BayesR and pGBLUP greatly outperformed GBLUP at sparse simulating settings...” in the results.
Line 205: Provide the range of predictive accuracy.
Response: Thanks for your suggestion. We have revised the sentence as “the predictive accuracy gain compared with GBLUP would be reduced from about 20% to 1%” in the results.
Discussion
The Discussion is fairly okay considering the fact that the study is relatively new.
Response: Thanks for your kind comment.
Conclusions
Line 316: Provide the cattle milk related QTL.
Response: Thanks for your comment. We have provided the cattle milk related QTL in the supplementary table 1.
References
All the 62 references are in the list.
Table S1
Why CHR X for gene MBNL3- WWC3?
Response: Thanks for your question. The genes MBNL3- WWC3 on chr X are located in the buffalo genomic region corresponding to the cattle milk traits QTLs. As in the section of 2.2 Animal resources and genomic information, the QTLs of 112 cattle milk related traits were downloaded from animal QTL database, the genes within 50 kb from the QTL regions were selected as associated genes to the trait. We aimed to improve genomic prediction of six buffalo milk traits by incorporating QTL information for cattle milk traits in this study.

Reviewer 2 Report
The authors have presented a new BLUP method of incorporating prior biological information to potentially improve prediction accuracy compared to standard GBLUP. While this method shows promising results in simulations, it seems less promising in the real world, where causative variants are not known completely or without error. Nevertheless, the method may prove useful over time as knowledge about true causative variants improves.
The name "PGBLUP" is already in use, where it refers to a version of GBLUP incorporating population data (e.g. Bian and Holland 2017 10.1038/hdy.2017.4). I recommend that the authors consider a different name for their method to avoid confusion.
It is not clear in the methods what the cattle QTL were used for: you say in section 2.2 that 396 genes were identified for prior information, but then go on to say that the simulation used randomly-selected causal regions. I'm guessing that they were used in the methods described in section 2.4 and results in section 3.2, but this is not at all clear. It would help if the results stated more clearly which parts came from the simulations and which came from real data.
In the real (non simulated) dataset, pGBLUP performed around the same as or slightly worse than the other two methods, probably because the real causative variants are not always known (the small sample sizes also won't help, as you state in the discussion); this would be particularly problematic when using QTL predicted in a different species, as it is likely that many buffalo QTL do not segregate in cattle. In light of this, is the pGBLUP actually likely to be improve prediction performance, as you state in your conclusions?
Relatedly, a proportion of the cattle causal sites will be fixed in buffalo, and therefore not contribute anything to the variance in that species. What would be the impact of including these, and how would you avoid/mitigate it?
Line 125: using a HWE threshold of p<10–4 to remove variants runs the risk of removing variants that are under selection (potentially interesting ones), as these may violate HWE. In fact, the structure of the cattle population as a whole, with small numbers of bulls contributing large numbers of progeny with the help of artificial insemination, already strongly violates HWE expectations, so setting such a strict filter may remove a lot of the causative variants. This is especially true when considering that the bulls are likely to be carrying good alleles for causative variants more often than not (because they are highly selected from the general population), which will bias the frequencies of these variants away from HWE. In my work, I typically use a threshold of 10–20 to try to avoid these problems, while still removing poorly genotyped or imputed loci.
Figure 2B: the gold dashed line is difficult to see against the green bars
Figure 3B could be improved by labelling the points by phenotype if this can be done clearly with the error bars
Line 317: I'm not sure how much help increasing SNP chip density will be, as other work in cattle has shown only small improvements (e.g. Su et al 2012 10.3168/jds.2012-5379). I believe that this is because QTL under recent strong selection tend to produce long runs of homozygosity that can be easily picked up by 50k density panels, so using higher density panels only adds QTL with smaller effects that are under weaker selection.
Author Response
Response of Comments and Suggestions for Reviewer2
The authors have presented a new BLUP method of incorporating prior biological information to potentially improve prediction accuracy compared to standard GBLUP. While this method shows promising results in simulations, it seems less promising in the real world, where causative variants are not known completely or without error. Nevertheless, the method may prove useful over time as knowledge about true causative variants improves.
Response: We thank the reviewer for the insightful comment.
The name "PGBLUP" is already in use, where it refers to a version of GBLUP incorporating population data (e.g. Bian and Holland 2017 10.1038/hdy.2017.4). I recommend that the authors consider a different name for their method to avoid confusion.
Response: Thanks for your kind recommendation. We noticed the "PGBLUP" method where “P” refers to “population” effects. In our approach, “p” refers to the “prior” biological/functional information. To distinguish our approach from “PGBLUP”, we have tried to use a lower-case “p” in the “pGBLUP”. We hope that “pGBLUP” can represent the characteristics of our approach and is suitable.
It is not clear in the methods what the cattle QTL were used for: you say in section 2.2 that 396 genes were identified for prior information, but then go on to say that the simulation used randomly-selected causal regions. I'm guessing that they were used in the methods described in section 2.4 and results in section 3.2, but this is not at all clear. It would help if the results stated more clearly which parts came from the simulations and which came from real data.
Response: We apologize for the confusion. In our simulation studies, we could set the genomic region to be causal or not, in other words, the causal regions were known. In the real data application of for genomic prediction of buffalo milk traits, the causal regions of buffalo milk traits were unknown, we hope to borrow information from the corresponding cow milk traits. To make it clear, we have added some details “In our simulation studies, the causal regions were simulated and known. Firstly, we assessed the performance of different methods...”, “In our real data application studies, according to the prior biological information of 396 cattle milk trait associated genes, 1279 SNPs within 10 kb from those genes were set in Z1 of pGBLUP model. The fe estimations of the selected 1279 SNPs in Z1 were shown in Figure 3A...” in the result part.
In the real (non simulated) dataset, pGBLUP performed around the same as or slightly worse than the other two methods, probably because the real causative variants are not always known (the small sample sizes also won't help, as you state in the discussion); this would be particularly problematic when using QTL predicted in a different species, as it is likely that many buffalo QTL do not segregate in cattle. In light of this, is the pGBLUP actually likely to be improve prediction performance, as you state in your conclusions? Relatedly, a proportion of the cattle causal sites will be fixed in buffalo, and therefore not contribute anything to the variance in that species. What would be the impact of including these, and how would you avoid/mitigate it?
Response: We thank the reviewer for the insightful comment. Intuitively, the biology process of milk production should be similar in cattle and buffalo, so we assumed that cattle and buffalo shared some common genes or genomic regions associated with milk traits. Specially, as we mentioned in our discussion, some known cattle milk related QTLs were also associated with buffalo milk traits (ref [25, 28, 56-59]). However, buffalo milk traits could be less artificially selected for some genes with large effects compared with cattle. We agree with the reviewer that some cattle causal sites may not contribute anything to the larger genetic variance in the buffalo. In such case, including information of less artificially selected genes in the Z1 will not benefit the pGBLUP as our polygenic simulation setting where pGBLUP and GBLUP had similar performance. In the real data application study, pGBLUP did not show its advantage in genomic prediction by integrating prior biological information, and three methods had similar performance with large standard errors of correlations. We thought one of reasons should be that we model all the cattle milk related QTL information in Z1, while not all SNPs in Z1 had large effects for buffalo milk trait. Although in simulations where pGBLUP had better performance in some ideal scenario, we also acknowledged and emphasized that genomic prediction for buffalo trait could be further improved with more precise prior biological information, as well as larger sample size, and higher density SNP chips in real data applications. In addition, we also weakened our statement as “We proposed a genomic prediction approach pGBLUP that had the potential to improve the genomic prediction performance by incorporating the proper prior biological information” in the conclusion part.
Line 125: using a HWE threshold of p<10–4 to remove variants runs the risk of removing variants that are under selection (potentially interesting ones), as these may violate HWE. In fact, the structure of the cattle population as a whole, with small numbers of bulls contributing large numbers of progeny with the help of artificial insemination, already strongly violates HWE expectations, so setting such a strict filter may remove a lot of the causative variants. This is especially true when considering that the bulls are likely to be carrying good alleles for causative variants more often than not (because they are highly selected from the general population), which will bias the frequencies of these variants away from HWE. In my work, I typically use a threshold of 10–20 to try to avoid these problems, while still removing poorly genotyped or imputed loci.
Response: We thank the reviewer for the insightful comment. We agree with reviewer that the HWE threshold of p<10–5 used in our study may remove some causal variants under strong selection. Here we applied a common used HWE threshold in GWAS and GS to filter the genotyping errors. Compared with cattle breeding, buffalo milk traits were less artificially selected, so we thought the HWE test might not remove lots of causal variants at the current threshold. In addition, we set all SNPs located in the potential QTL regions in the Z1. Even though the causal variant might be filtered in HWE test, its nearby and correlated variants could still provide the information for prediction. We did not re-analyze the data using a new threshold of HWE test, however we discussed that the HWE p threshold may affect our prediction “In addition, we used a relatively loose threshold of HWE test (p-value < 10−5) to remove the variants due to the genotyping errors, which was also likely to remove the causal variants under selection, thus affecting the performance of genomic prediction.” in the discussion part.
Figure 2B: the gold dashed line is difficult to see against the green bars
Response: Thanks for your suggestion. We have updated the figures 2B and used the bold black dash lines.
Figure 3B could be improved by labelling the points by phenotype if this can be done clearly with the error bars
Response: Thanks for your suggestion. It’s a little difficult to label the point by phenotype, because some points were very close. Actually, we provided the correlations and standard errors in the Table 1.
Line 317: I'm not sure how much help increasing SNP chip density will be, as other work in cattle has shown only small improvements (e.g. Su et al 2012 10.3168/jds.2012-5379). I believe that this is because QTL under recent strong selection tend to produce long runs of homozygosity that can be easily picked up by 50k density panels, so using higher density panels only adds QTL with smaller effects that are under weaker selection.
Response: We thank the reviewer for the insightful comment. We agree with reviewer that QTL under recent strong selection tended to produce long runs of homozygosity that can be easily picked up by 50k density panels in Cattle breeding. However, such benchmark comparison has not been fully studied in buffalo as we know. As our reply in previous comment, we thought that buffalo milk traits were less artificially selected compared with cattle. The high density SNP panel and imputation could aid the findings of causal variants or some variants with large effects, thus having the potential to improve the genomic prediction in buffalo.